# Glutathione-Related Enzymes and Proteins: A Review

**DOI:** 10.3390/molecules28031447

**Published:** 2023-02-02

**Authors:** Janka Vašková, Ladislav Kočan, Ladislav Vaško, Pál Perjési

**Affiliations:** 1Department of Medical and Clinical Biochemistry, Faculty of Medicine, Pavol Jozef Šafárik University in Košice, 040 11 Košice, Slovakia; 2Clinic of Anaesthesiology and Intensive Care Medicine, East Slovak Institute of Cardiovascular Disease, 040 11 Košice, Slovakia; 3Institute of Pharmaceutical Chemistry, University of Pécs, 7600 Pécs, Hungary

**Keywords:** cell, redox homeostasis, glutathione, glutathionylation, glutathione system, glutathione enzyme

## Abstract

The tripeptide glutathione is found in all eukaryotic cells, and due to the compartmentalization of biochemical processes, its synthesis takes place exclusively in the cytosol. At the same time, its functions depend on its transport to/from organelles and interorgan transport, in which the liver plays a central role. Glutathione is determined as a marker of the redox state in many diseases, aging processes, and cell death resulting from its properties and reactivity. It also uses other enzymes and proteins, which enables it to engage and regulate various cell functions. This paper approximates the role of these systems in redox and detoxification reactions such as conjugation reactions of glutathione-S-transferases, glyoxylases, reduction of peroxides through thiol peroxidases (glutathione peroxidases, peroxiredoxins) and thiol–disulfide exchange reactions catalyzed by glutaredoxins.

## 1. Introduction

Glutathione (GSH) was first isolated in 1888 by De-Rey-Pailhade. He named the substance *phylothion*, the Greek expression for sulfur loving [1]. Its structure was controversial for several years. Initially, it was described as a sulfur-containing dipeptide [2]. Later the structure was refined, demonstrating that the substance is a tripeptide, γ-Glu-Cys-Gly [3,4,5]. Other related compounds, such as γ-Glu-Cys-Gly-spermidine and (γ-Glu-Cys)*_n_*-Gly in *E. coli* and plants, were also described [6]. The thiol group of the cysteine residue enables GSH to function as both a reducing agent and a nucleophilic center [7]. Glutathione occurs in two free forms: the reduced (GSH) thiol and the oxidized (GSSG) disulfide forms (Figure 1). In addition, it can be bound to proteins and other thiols, affecting their activity. In its reduced and oxidized forms (GSH, GSSG), glutathione is ubiquitous in mammalian cells ranging in 1–10 mM concentrations [8]. Under physiological conditions, more than 98% of total GSH occurs in the reduced form [9,10]. It is an essential antioxidant against reactive oxygen and nitrogen species [11]. The compound plays a critical role in maintaining the redox homeostasis of the cells and in cell cycle regulation, apoptosis, immunological defense, and pathological abnormalities [8]. Furthermore, it is one of the endogenous substances involved in the metabolism of endogenous (e.g., estrogens, leukotrienes, prostaglandins) and exogenous compounds (e.g., drugs, non-energy-producing xenobiotics) [12]. These latter transformations could be the molecular basis for eliminating foreign substances from the body. In this review, the role of glutathione and glutathione-dependent enzymes in the maintenance of redox homeostasis is summarized.

## 2. Glutathione

Together with glutaredoxins (Grx), GSH acts to reduce disulfide bonds and is, in turn, oxidized to glutathione disulfide (GSSG), which is reduced by NADPH-dependent glutathione reductase. The GSH/GSSG, NADPH/NADP^+^, Grx-SH/Grx-SS, and Trx-SH/Trx-SS are the most important redox couples in maintaining cellular redox homeostasis [13]. The standard apparent redox potential (*E’^o^*) of GSH is −288 mV (pH 7, 298.15 K, 0.25 M ionic strength), which is well between the most negative H^+^/H_2_ (−423 mV) and the most positive, O_2_/H_2_O (+849 mV) redox couples [14]. Accordingly, the GSH/GSSG redox couple can readily interact with most physiologically relevant redox couples, undergoing reversible oxidation or reduction [7].

Given the availability of glutathione in the cells, the reactions of protein thiols are mediated by multiple enzymes and enzyme systems, thus allowing it to participate in the abovementioned functions and regulatory pathways. Among them are glutaredoxins, which are central in the response against oxidative stress as the biological activity of many proteins are modified by the formation of GSH-mixed disulfides. Furthermore, other redox-maintaining enzymes such as glutathione peroxidases, and detoxification enzymes, glyoxylases, are closely related to carbohydrate metabolism [15,16]. Thus, the involvement of glutathione and its activity in the cell represents a wide range of biological and biochemical processes. The consequence of its deficiency results in increased stress conditions, which is the basis of the pathophysiology of many organ or tissue-specific diseases such as inflammation, virus infections (HIV), sickle cell anemia, cancer, diabetes, heart attack, stroke, liver disease, cystic fibrosis, Alzheimer’s, and Parkinson’s disease [17,18]. 

### 2.1. The Role of the Liver in Glutathione Synthesis and Distribution

Synthesis of GSH occurs in the cytoplasm in all cells in two subsequent ATP-dependent reactions catalyzed by glutamate-cysteine ligase and GSH synthetase, from where it is transported to other organelles and extracellular space [8,19]. Glycine, glutamate, and cysteine as nonessential amino acids can be obtained from dietary sources or synthesis.

The liver removes a significant amount of resorbed cysteine from the portal vein [20]. However, cysteine can be synthesized by methionine transsulfuration in the liver [21]. The liver is responsible for the metabolism of up to half of the daily methionine intake, predisposing the liver to almost exclusive transsulfuration activity and being the most important in interorgan GSH homeostasis [22]. Thus, a considerable amount of GSH is produced by the liver and released into plasma and bile [22]. Rat liver cytosolic GSH has a half-life of 2–3 hours [8], and the daily turnover for GSH is estimated to be higher than cysteine turnover in the body protein pool, around 40 mmol per day [21]. Transsulfuration is not present in the fetus, newborn infants, or patients with cirrhosis [23]. Cirrhosis causes a decrease in methionine adenosyltransferase activity following a reduction in S-adenosylmethionine production and lower effectivity of the transsulfuration pathway [24]. Glutathione concentration within extracellular fluids and blood plasma reaches only several μM; however, in some extracellular fluids, such as lung lining fluid, 100–400 μM levels have been detected [25,26].

### 2.2. Cell Uptake and Metabolism of Glutathione

To date, two mechanisms of glutathione uptake into mammalian cells are known [19]. The most common one is primarily associated with the activity of γ-glutamyl transpeptidase (GGT) (Figure 2). GGT is localized to the cell surface and cleaves only extracellular substrates, GSH, and oxidized GSH (GSSG), its most abundant ones. The amide bond between the glutamine γ-carboxyl and the cysteine amino units does not allow cleavage of GSH by cellular and circulating serum peptidases [27]. It is hydrolyzed by the γ-glutamyltranspeptidase (GGT) to glutamate and Cys-Gly. Cys-Gly can be cleaved by membrane-bound dipeptidases (MDBs) or intracellular Cys-Gly peptidases. Cellular uptake of Cys-Gly or the individual Cys, Gly, and glutamate units serve as precursors for intracellular GSH synthesis. GGT is expressed on the luminal surface of excretive and absorptive cells that line glands and ducts throughout the body, with the highest level of GGT activity in the kidney and pancreas ducts [28]. It is nearly absent, however, from the hepatocytes and cardiac myocytes [7]. The absence of GGT activity on the apical surface of the kidney’s proximal tubules by genetic disorder results in glutathionuria [29,30].

GGT has multiple functions, including catalytic transfer of γ-glutamyl groups to amino acids and short peptides, hydrolysis of GSH to glutamyl moiety and cysteinyl glycine, and catabolism of GSH conjugates [31]. GGT allows hydrolysis of a broad range of γ-glutamyl amides and transpeptidation of amino acids or dipeptides [32]. So GSH, its *S*-conjugates, GSSG, γ-glutamyl di- or tripeptides, glutamine, l-α-methyl derivatives of γ-glutamyl amides, various lipid-derived mediators (e.g., leukotriene C4), geranylgeranyl, poly-γ-glutamyl derivatives serve as substrates of GGT [33,34,35,36]. Many tumor cells express GGT on their entire cell surface and can therefore cleave GSH not only in the ductal but also in interstitial fluid and blood [37]. GGT expression provides tumor cells with an additional source of cysteine and cystine from the breakdown of extracellular GSH and GSSG [38]. 

Besides the GGT pathway, there is evidence of Na^+^-dependent and Na^+^-independent glutathione transport systems for glutathione cell uptake expressed in the renal basolateral membrane [38,39], the small intestine [40], and the brain [41]. In the renal basolateral membrane, two Na^+^-independent Organic Anion Transport systems (OAT1 and OAT3) [42] and the Na^+^-dependent dicarboxylate carriers are the most important organizations [43,44]. On the other hand, the plasma membrane glutathione efflux can be facilitated by specifically or ubiquitously expressed membrane proteins and anion channels such as multidrug resistance-associated proteins (MRP1-5), Cystic Fibrosis Transmembrane Conductance Regulator (CFTR), Arginine/Ornithine Transport ATP-binding Proteins (OATP 1,2), and ATP-Binding Cassette superfamily G member 2 (ABCG2) [19].

### 2.3. Intracellular Distribution and Functions of GSH

Within the cell, there are three main glutathione pools. The cytosol (80–85%), the mitochondria (10–15%), and the endoplasmic reticulum [45,46,47]. Studies by Birk et al. and Montero et al. [48,49] pointed out that the total glutathione content in the lumen of the endoplasmic reticulum even exceeds the entire cellular glutathione content. GSH and GSSG concentrations depend on the subcellular compartment, the cell type, and the organism. Accordingly, the redox potential of the GSSG/2GSH system varies from tissue to tissue, from organism to organism. This relies on the proportion of GSH and GSSG and the total concentration of glutathione, which is quite challenging to estimate their actual concentration and ratio in vivo [50,51]. For example, taking the local pH and GSSG/2GSH ratios into consideration, cytosolic *E*_pH7.0_ = −289 mV (or even lower), mitochondrial matrix *E*_pH7.0_ = −296 mV (or even lower), and human plasma *E*_pH7.4_ = −118 mV half-cell reduction potentials (*E*_hc_) have been estimated [52]. Furthermore, a correlation has been found between the cell cycle, the condition of the cell (stressed, apoptotic, etc.), and the GSSG/2GSH ratio. For instance, in cell proliferation (*E*_hc_ = ~−240 mV), in cell differentiation (*E*_hc_ = ~−200 mV), and in apoptosis (*E*_hc_ = ~−170 mV), which can be applicable for a better understanding of oxidative stress [13,53]. Van’t Erve et al. [54] found that GSSG/2GSH levels and reduction potential in erythrocytes reflect genetic differences between individuals.

Cytoplasmic glutathione levels impact glutathione diffusion through nuclear pore complexes [55], playing a role in oxidative signaling during proliferation, epigenetic control of histone activity, and the cell cycle control, mainly in the S + G_2_/M phase [56,57]. ATP-dependent transporters have also been reported to import glutathione into the nucleus [58]. 

Glutathione synthesis occurs only in the cytosol; thus, the mitochondrial pool is supplied by GSH transport and maintained by reducing its oxidized form via the activity of glutathione reductase. Glutathione passes the mitochondrial outer membrane through the mitochondrial porin, a voltage-dependent anion channel (VDAC). As a negatively charged molecule, glutathione cannot diffuse through the mitochondrial inner membrane. Its transport into the mitochondrial matrix is either active or provided in exchange for another anion [7]. Six of the eight anion carriers have the potential for GSH import through the inner membrane into mitochondria. Monocarboxylate, dicarboxylate (DIC), 2-oxoglutarate (OGC), tricarboxylate (or citrate), glutamate-hydroxide, glutamate-aspartate transporters involved in the transport of GSH also provide intermediates of the Krebs cycle and the gluconeogenesis pathway [59]. DIC and OGC were identified as major GSH transporters, although at the expense of Krebs cycle intermediates [60]. Around 70–80% of GSH transport could be associated with DIC and OGC activity in the kidney, but only about 45–50% of liver mitochondria [61]. DIC imparts malate (malonate or succinate) in exchange for phosphate, sulfate, and thiosulfate. Malate conversion into oxalacetate, followed by the formation of phosphoenolpyruvate, is limited for gluconeogenesis in the cytosol. Reduction in DIC expression leads to decreased glutathione levels and impaired complex I activity [62]. OGC transfers 2-oxoglutarate substituting dicarboxylate [63], thus regulating respiration and glycolysis. While succinate from the matrix side increases the affinity of OGC to malate, substrates such as phenyl succinate, pyridoxal phosphate, retinoic acid, and ethanol cause inhibition of OCG. Reduced activity of OCG leads to lower energy production, increased oxidative stress, and it could be the basis of liver or nervous tissue diseases [64,65,66]. GSSG is not transported out from mitochondria [67].

The endoplasmic reticulum offers a unique setting concerning GSH homeostasis. It contains the thiol oxidase Ero1, which catalyzes the formation of disulfides transmitted to folding substrates via protein disulfide isomerase (Pdi1). Both reduced and oxidized forms of glutathione are transported into the endoplasmic reticulum at different rates, with a preference for the reduced form [68]. Since GSH is oxidized but not reduced in the ER, GSH must be imported into the ER, while GSSG is exported to the cytosol [69]. A study by Ponsero et al. [70] brought up the finding of facilitated diffusion of GSH through the Sec61 protein translocation complex. In the sarcoplasmic reticulum, ryanodine receptor calcium channel type 1 (RyR1) was suggested to play an important role [71]. However, Bachhawat et al. [19] pointed out that this might result from the S-glutathionylation of several cysteine residues within the RyR1 molecule. To maintain GSH homeostasis, part of GSSG is transported to the cytosol through vesicular transport [72]. Most GSSG reacts with proteins or protein disulfide isomerase involved in oxidative protein folding [64]. A lower GSH:GSSG ratio results in more oxidizing conditions (−240 mV) [70] in the endoplasmic reticulum allowing protein disulfide formation.

GSH plays an essential role not only in the peripheral tissues but in the central nervous system (CNS) as well. Brain tissues are rich in unsaturated fatty acids. Due to their relatively low levels of antioxidants or antioxidant enzymes, they are rather sensitive to oxidative damage. The most important small molecular CNS antioxidants are GSH, ascorbic acid (vitamin C), and α-tocopherol (vitamin E) [73]. Among these antioxidants, GSH seems to be the determining agent because it is selectively decreased in the brains of patients with these neurodegenerative diseases (e.g., Parkinson’s disease, Alzheimer’s disease, and Amyotrophic lateral sclerosis) [74]. Therefore, regulating the redox state by intracellular GSH is crucial for maintaining cellular functions under physiological and pathological conditions.

In the central nervous system, besides the functional neurons, there are several other types of cells for the nervous system to function properly. This is where a set of glial cells intervene, which make up 25–50% of the nerve mass [75]. The most common type of glial cells in the CNS are the astrocytes and the microglia. Synthesis of GSH occurs both in the neurons and the glial cells. In an early work by Rice and Russo-Menna (1998) [76], GSH levels of glutathione in neurons and glia were reported to be 2.5 nM and 3.8 mM, respectively. The authors found that ascorbate predominates in neurons (10 mM), whereas GSH is slightly predominant in glia. According to the above, GSH supplementation seems promising for treating patients with neurodegenerative diseases.

### 2.4. Acid–Base Properties

The acid–base properties of glutathione (GSH) have long been the focus of scientific interest. It has three acidic (thiol, glycinyl carboxyl, glutamyl carboxyl) and one basic (amino) functional group. Accordingly, in an aqueous solution, glutathione can exist in four different macroscopic protonation states:$$L^{3-}\overset{K_{1}}{↔}{\mathrm{HL}}^{2-}\overset{K_{2}}{↔}H_{2}L^{-}\overset{K_{3}}{↔}H_{3}L\overset{K_{4}}{↔}H_{4}L^{+}$$
where L^3−^ is the fully deprotonated, H_4_L^+^ is the fully protonated GSH molecule.

Since the HL^2−^ and the H_3_L forms have four protonation isomers (microspecies) each, and the H_2_L^−^ form has six microspecies, the molecule has sixteen different protonation states (microspecies) altogether [77].

The micro and sub-micro protonation constants characterize the acid–base properties at the submolecular level [78]. These constants allow quantification of the proton binding capacity of submolecular basic units when the protonation states of all other sites are defined in the molecule [79]. Group constants are special micro constants when the rest of the groups in the molecule are far enough apart, and their protonation does not affect the basicity of the group [80]. The rotational state of the flexible parts of the molecules is defined by the sub-micro constants when protonation occurs [81]. The correct characterization of the basicity of the sites of protonation of multidentate ligands can be conducted using the micro and sub-micro constants. In addition, this group of constants is used to measure the concentration of different protonation forms, of which the principal form is not always the reactive form in chemical and biological processes. [82,83,84,85,86]. The macroscopic protonation constants (K1-K4) determined by ^1^H NMR-pH titrations were as follows: logK1 9.65; logK2 8.78; logK3 3.52; and logK4 2.22 [77].

The obtained values were found to be very similar to those determined in earlier works of Pirie and Pinhey [87] (9.62, 8.66, 3.53, 2.12), Li et al. [88] (9.65, 8.75, 3.59), and Martin and Edsall [89] (9.62, 8.74). The results demonstrated that the first and the second protonation constants were predominated by the overlapping protonation of the amino and the thiolate site, the amino being typically more favored. The carboxylate groups also protonated in an overlapping fashion, the glycinyl carboxylate being more basic. It is worth mentioning that the protonated amino group makes the inherently more basic glutamyl carboxylate more acidic [77].

It is important to note that the physico-chemical properties (e.g., complex formation, nucleophilic reactivity, redox properties) and biological functions of glutathione could be significantly different at different protonation states (i.e., in solutions with different pH values) [90,91,92,93] and its redox behavior [94,95]. Furthermore, ionic strength and the nature of ionic media also affect the acid–base characteristics of glutathione [96].

### 2.5. Antioxidant Properties

The pKa value of GSH (ranging from 8.6 to 8.8 [87,88,89] results in low thiol reactivity in the cellular environment [97]. Still, high GSH concentrations enable some reducing activities against oxidizing agents in the cell [98]. GSH, for example, can reduce H_2_O_2_, resulting in GSSG and water [99]. The rate of reaction depends on the cellular GSH level and the ratio of GSH to H_2_O_2_ concentrations [100]. Recently, Zinatullina et al. [101] confirmed that the oxidation of GSH is accompanied by radical formation. GSH reacts with the majority of free radicals generating thiyl radicals. Consecutive reactions of the radicals with a thiolate anion and molecular oxygen lead to disulfide and superoxide radicals formation [102]. Furthermore, γ-glutamylcysteine, a GSH precursor, was found to decompose H_2_O_2_ similarly to glutathione peroxidase-1 [103].

Glutathione exists in 100 μM concentrations as glutathione persulfide (GSSH) [104], the latter exhibiting higher activities due to its higher nucleophilic power than GSH [105]. Under specific conditions, GSSH reacts with H_2_O_2_, while GSH does not [106]. Furthermore, its reactions with one-electron oxidants are faster than similar reactions of thiols [107]. GSSH are intermediates in the synthesis of iron-sulfur clusters and mitochondrial H_2_S oxidation [108,109,110]. GSH can react with HS^−^ catalyzed by sulfide quinone oxidoreductase or thiosulfate sulfurtransferase, forming GSSH, which can reduce oxidized thioredoxin. Single-domain sulfurtransferase (TSTD1, known as rhodanese) and mercapto pyruvate sulfurtransferase can also directly transfer sulfides to GSH and the thioredoxin antioxidant systems [111]. Mutations in persulfite dioxygenase, oxidizing GSSH to sulfite and GSH, are bases for autosomal-recessive inherited ethylmalonic encephalopathy [112].

### 2.6. Redox Signaling Properties

Signaling is the process that makes cells capable of reacting to the change in their environment (intercellular signaling) or their homeostasis (intracellular). The initial step of the process is the interaction of the signaling particles (ligands) with the target molecule (receptor). The well-known signaling mechanisms involve protein–protein interactions, allosteric changes induced by the binding of ligands, proteolytic processing, and chemical modifications such as acylation, acetylation, alkylation, and phosphorylation of proteins. On the contrary, redox signaling is the transduction of signals based on the transfer of electrons. Redox signaling involves a broad spectrum of pathways involving free radicals, redox-active metals (e.g., iron, copper), or reductive equivalents [74]. Here only those pathways are mentioned that are based on a modification of signaling proteins through the modification of one amino acid, cysteine.

The physiological level of hydrogen peroxide) (H_2_O_2_) and nitric oxide (^∙^NO) can selectively react with the thiol function of the cysteinyl residues at the active site of the proteins (receptors, enzymes, transporters, etc.). Accordingly, the receptor-mediated stimulation of the H_2_O_2_ and ^·^NO production are part of normal physiology; this is especially true for the longer-lived H_2_O_2_. [113]. However, overproduction of these and related species (ROS and RNS) lead to irreversible oxidation of the thiol residues and impairs cellular protein functions [114,115]. The GSSG/2GSH redox system is fundamental in the cells and, together with other redox-active couples (including NADPH/NADP+, Trx-SH/Trx-SS), regulates and maintains the appropriate cellular redox status. For example, the GSSG/2GSH half-cell reduction potential differed in cell proliferation, differentiation, and apoptosis [13,53]. Thus, changes in the GSSG/2GSH ratio are fundamental in controlling signal transduction that supports cell cycle regulation and other cellular processes [55].

The functions and activities of GSH as the main regulator of cellular redox status and redox signal transduction have been reviewed [17,116,117,118,119]. GSH acts protectively against oxidative stress by reacting directly with ^·^NO, superoxide anion radical (O_2_**^·^**^−^), H_2_O_2_, hydroxyl radical (^·^OH), peroxinitrite anion (ONOO^−^), and the lipid peroxidation product 4-hydroxy-2-nonenal (4-HNE) [116,117]. Such reactions directly modify the cellular GSSG/2GSH half-cell potential, a physiological signaling event. Furthermore, changing the GSH level results in a selective change in the activity of the thioredoxin/glutathione systems [118], the glutaredoxin/glutathione system [119], and the activity of some GST isoforms. The latter protein family is involved not only in the metabolism of xenobiotics but also of endogenous compounds which play critical roles in regulating signaling pathways [120,121,122].

### 2.7. Reactions with Electrophilic Xenobiotics

Glutathione-S-transferases (GST) lower the pKa of GSH thiol under 6, enhancing rates of nucleophilic addition and substitution reactions with electrophylic xenobiotics (Figure 2). These reactions are examples of Phase II bioconjugation reactions, most of which result in reduced toxic effects of the parent compounds or their metabolites [98,123]. Other enzymes/enzyme systems, e.g., selenium-containing glutathione peroxidases (GPx) or peroxiredoxins (Prdx), use GSH to reduce various peroxides and hydroperoxides. Glyoxalase (Glo) performs conjugation of GSH with the glycolysis byproduct methylglyoxal to form (S)-lactoylglutathione (Figure 2). Moreover, glyoxalase II (Glo-2) catalyzes S-glutathionylation using (S)-lactoylglutathione [124].

## 3. The Glutathione Peroxidase System

The glutathione or glutathione peroxidase system consists of glutathione peroxidase (GPx) and glutathione reductase (GR). In the decomposition reaction of H_2_O_2_ or other organic peroxides (HOOR), two molecules of GSH reduce the substrate to H_2_O or the corresponding alcohol (HOR) and restore the enzyme forming GSSG with concomitant formation of GSSG and H_2_O.
2 GSH + HOOR → GSSG + HOR + H_2_O(1)

GSSG can be excreted from the cell or recycled by GR using the reducing power of NADPH (Figure 3). NADPH arises in two reactions of the pentose phosphate pathway, which is the most potent source of it. However, NADPH can also be formed directly in the mitochondria by NAD(P)^+^ transhydrogenase, mitochondrial/cytosolic NADP-dependent isocitrate dehydrogenase, or cytosolic malate dehydrogenase [125].

GR is a homodimeric flavoprotein consisting of 52 kD monomers. Except for synthesis, the activity of GR represents a second source of GSH in the cytosol and some organelles, such as mitochondria. Although inhibition of GR has been reported to cause a depletion of GSH and accumulation of GSSG [126], a comprehensive study of the GR and the cellular thiol redox system is missing [127]. Inhibition of the enzyme has also been related to the toxicity of various chemicals and metals [128,129].

The term glutathione peroxidase (GPx) describes only a small subgroup of the peroxidases [130], which belong to a group of phylogenetically related enzymes. GPx 1–4 are selenoproteins with selenocysteine (SeCys) in the catalytic center. GPx6 is a human selenoprotein [131]. Their important antioxidant function was shown in various places and cell structures: GPx1 is ubiquitous in the cytosol and mitochondria, GPx2 in the intestinal epithelium, and GPx3 in the plasma; all three work in the aqueous phase reducing H_2_O_2_ and free fatty acid peroxides [131]. GPx4 protects mainly membranes by reducing phospholipid and cholesterol peroxides [131,132]. Gpx5, which contains cysteine instead of Se in the active center, is a secretory enzyme of the epididymis. GPx6 is a human selenoprotein and is formed by the olfactory epithelium. GPx7 and GPx8 are also CysGPx with low peroxidase activity. GPx1, 2, 3, 4, 5, and 6 are homotetramers, which could determine their specificity for hydrogen peroxide. GPx4, 7, and 8 are monomers. This structure probably enables the reaction with more complex lipid hydroperoxides, but this has been proven only for GPx4 [132]. The catalytic center of GPx was first characterized as a triad consisting of SeCys or Cys, Gln, and Trp. It was later found to be a tetrad with Asn. A conservative feature for these GPx is the presence of a second or even a third cysteine residue.

The reaction mechanism differs between individual GPx isoforms, whose activity requires GSH. In general, they do not form a ternary complex between the enzyme, hydroperoxide, and GSH, but the reaction has a concomitant oxidation and a reduction part. In the oxidation part, deprotonation takes place in the same way. The side chains of the Glu, Try, and Asp residues form a highly nucleophilic region in the enzyme’s active center, where oxidation of the active site selenocysteine (RSeH) or cysteine (RSH) occurs after binding the peroxide. This reaction results in the formation of a selenenic acid (RSeOH) derivative. The selenenic acid is then converted back to the selenol (RseH) by a two-step process that begins with a reaction with GSH to form the GS-SeR and water. A second GSH molecule reduces the GS-SeR intermediate back to the selenol, releasing GS-SG as the byproduct [52,132]. A simplified representation (with H_2_O_2_ as a substrate) is shown below:RSeH + H_2_O_2_ → RSeOH + H_2_O(2)
RSeOH + GSH → GS-SeR + H_2_O(3)
GS-SeR + GSH → GS-SG + RSeH(4)

Glutathione reductase then reduces the oxidized glutathione to complete the cycle:GS-SG + NADPH + H^+^ → 2 GSH + NADP^+^(5)

Selenium deficiency results in increased GSH synthesis in the liver with accompanying release to the plasma [133]. Increased plasma GSH led to cysteine depletion, impaired protein synthesis, decreased GPx, and increased GST activities [134]. Usually, GPx requires GSH in millimolar concentrations in the intracellular space, and plasma GSH reaches micromolar concentrations, which questions GPx’s antioxidant function [135]. However, within the cell, in the cytosol and mitochondria, the GPx system appears to be very efficient in the elimination of H_2_O_2_ due to the low (100–200 μM) K_m_ value of the enzyme [136] and the range of substrates [137]. Mimicking GSH, γ-glutamylcysteine can be used by GPx1 as a cofactor [103].

## 4. Glutaredoxins (Grx)

The thiol oxidoreductase glutaredoxins (Grx) are small proteins reducing various protein disulfides (PrSSPr) and GSH-protein mixed disulfides (PrSSG), where the electron donor is glutathione [138]. Grxs catalyze glutathionylation, post-transcriptional modifications, and the disulfide exchange between GSSG and protein thiols (PrSH) [139] (Figure 3). Grx-catalyzed (de)glutathionylation is an important event in signal transductions and serves as the primary protective mechanism against the irreversible oxidation of cysteine residues [115]. As mentioned above, the standard cell potential changes depending on the environment and the cell itself. Cell proliferation occurs at approximately −240 mV, differentiation at about −200 mV, and apoptosis at around −170 mV [55]. Changes in the GSH/GSSG redox potential can be sensed by Grxs, which operate as GSH-dependent reductases at about −240 mV and GSSG-dependent oxidases at about −170 mV [140].

Grxs are characterized by their active site motif. Dithiol-type Grx (class I) enzymes have a Cys-Pro-Tyr-Cys active site, while monothiol Grx (class II) enzymes do not contain a thiol at the C-terminus of the active site (Cys-Gly-Phe-Ser). Dithiol Grxs and monothiol Grxs with one Grx domain are found in all living organisms. Multi-domain monothiol Grxs (PICOTs, PKC-interacting cousin of thioredoxin) are present in eukaryotic cells. These contain an N-terminal Trx-like domain and three C-terminal monothiol Grxs domains [141]. Two other regions were recognized near the active site, the Grx characteristic motif GG and the TVP, which are involved in binding GSH [142].

### 4.1. Glutathionylation

Glutathionylation involves the reversible attachment of glutathione to cysteine residues in target proteins. Conditions of elevated oxidative stress increase the levels of protein glutathionylation. The glutathionylation/deglutathionylation cycle is viewed as a process that acts primarily against ROS/RNS via reducing aberrant cysteine modifications and thereby preventing the formation of damaging irreversible cysteine modifications.

There are three pathways of glutathionylation. (a) The thiol-disulfide exchange between GSSG and PrSH is accomplished at a low GSH:GSSH ratio. The reactivity of PrSH depends on the thiol pKa [143]. (b) The oxidation of the PrSH yields a thiyl radical (RS**^·^**), which reacts with the deprotonated form of glutathione (GS**^−^**), forming a mixed disulfide radical (RSSG**^·−^**). After the loss of an electron, a mixed disulfide (RSSG) and a superoxide anion radical (O_2_**^·^^−^**) are formed [144]. (c) Mixed disulfides can also be formed with low molecular weight thiols with indistinct biological relevance. As Lushchak [60] discussed, inhibition of glutathione reductase, phosphofructokinase, fatty acid synthase, or activation of fructose-1,6-bisphosphatase by CoASSG was shown.

Cysteine residues of proteins with a low pKa are targets for redox modulation under oxidative or nitrosative stress conditions. The primary products of these oxidative transformations are the respective thiyl radicals (PrS**^.^**). These reactive intermediates can react with glutathione (GSH) to form stable glutathionylated protein disulfides (PrSSG) to prevent their further oxidation with molecular oxygen. The protected protein thiol can be regenerated by the deglutathionylation process (e.g., through a reaction with another GSH molecule). Under oxidative stress, the thiyl radical can be further oxidized to form sulfenic (RSOH), sulfinic (RSO_2_H), or sulfonic acid (RSO_3_H) derivatives of the proteins. Both sulfenic and sulfinic acids of proteins can be reduced by Trx and sulfiredoxin, respectively [145,146,147]. In contrast, sulfonic acid cannot be reduced. Both sulfenic and sulfinic acids of proteins can be conjugated to GSH to form S-glutathionylated proteins via glutathione S-transferases (GSTs), Grx, or nonenzymatically. Glutathionylation was referenced to cytoskeletal proteins, metabolic, redox enzymes, cyclophilin, stress proteins, nucleophosmin, transgelin, galectin, and fatty acid binding protein [148], affecting their activity either in activation or decrease.

### 4.2. Deglutathionylation

Deglutathionylation undergoes cleavage of the disulfide linkage of the glutathionylated protein with another GSH molecule (Figure 3). The reaction can proceed (a) either in a mixed disulfide intermediate with an N-terminal thiol active site; (b) in a mixed disulfide intermediate by the attack of a second GSH molecule; or (c) by non-covalent binding of the thiol function of both an *N*-terminal thiol active site and GSH-coordinating metal cofactor in the [Fe-S] binding Grx subgroup [142]. The motif in the active site and the type of disulfide bond in the target protein are decisive for the reaction mechanism [149]. In the reaction mechanism of monothiol Grxs, the reduction of glutathionylated proteins (PrSSG) begins with a nucleophilic attack of the *N*-terminal cysteine. As a result, glutathionylated Grx and reduced substrate protein are released. The Grx-SG intermediate is cleaved by a GSH molecule, resulting in reduced Grx and GSSG, which is subsequently reduced by GR [150] (Figure 3). In the mechanism of dithiol Grxs, the reduction of PSSG and mixed disulfides begins with a nucleophilic attack of the *N*-terminal cysteine, but GSH is released. The Grx-protein intermediate is reduced by the second C-terminal active cysteine of Grx, forming oxidized Grx and reduced protein [151,152]. Dithiol Grx can also use monothiol mechanisms. However, both mechanisms are critically dependent on the availability of reduced GSH [153].

Apart from oxidoreductase activity, both classes of Grx proteins can bind [Fe-S] clusters. Class II enzymes are essential in the processes of regulation of Fe metabolism. Their function depends on the [Fe-S] binding capacity and not on the reductase activity [154]. In addition, Grxs have dehydroascorbate reductase and transhydrogenase activity, catalyzing denitrosylation and partial cystine conversion [155].

Monothiol Grxs (Grx3 and Grx5) form an iron–sulfur complex. Both isoforms can transfer iron to specific proteins. However, monothiol Grxs cannot deglutathionylate target proteins [156]. Grx3, localized in the cytosol, has a unique domain structure consisting of an *N*-terminal Trx-homology domain [141,157]. The first discovered function of Grx3 was related to that of protein kinase C theta, and in T-cells, Grx3 colocalizes with it, hence the name PICOT [157]. Since Grx3 is expressed in a wide variety of organs and tissues, it has been proposed as a redox sensor in signal transduction in response to reactive oxygen and nitrogen species [158]. Nuclear Grx3 has a role in the epigenetic regulation of chromatin by regulating the methylation of myelin transcription factor 1 and cell proliferation [159,160]. Grx5 participates in the biogenesis of [4Fe–4S] clusters by interacting with ISCA1 of the mitochondrial homolog of the iron–sulfur cluster assembly and ISCA2 of the cytosolic iron cluster [161,162]. Grx5 forms a cluster in the cytosol with a family of BolA-like proteins (regulatory DNA-binding proteins) for the maturation of iron–sulfur proteins [163].

Grx1 and Grx2 are dithiol Grxs. Most human Grx1 is found in the cytosol, less in the nucleus [164] and the mitochondrial intermembrane space [165]. Grx1, unlike Trx, is not an essential protein [98]. Grx1 activity depends on the redox state of the cells, especially the GSH/GSSG ratio [166]. In addition to deglutathionylation activity, Grx1 has also been able to denitrosylate protein Cys-NOs and prevent the pro-apoptotic effect of nitric oxide in tumor cell lines and cardiomyocytes [167,168]. Grx2 is about 20 times less abundant than Grx1 [169]. Depending on gene splicing, it is localized in mitochondria, cytosol, or nucleus [170]. Like Grx1, it catalyzes the reduction of disulfides mixed with GSH with a higher affinity but with a lower turnover rate [171]. However, these two proteins behave differently in response to an oxidative environment. While Grx1 is inhibited when other structural cysteine residues are oxidatively modified [154], Grx2 is activated. The different response to oxidative conditions is due to the ability of Grx2 to form [Fe–S] clusters [172]. The [Fe–S] clusters act as sensors for Grx2 activity under oxidative conditions [154]. Outside the active site, two cysteines form a [2Fe–2S]-bridged dimer that is enzymatically inactive. Oxidative stress increases GSSG concentration and reduces the availability of GSH for coordination of the [Fe–S] complex, leading to cluster degradation and formation of enzymatically active Grx2 monomers [154]. Grx2 can cycle and accept electrons from thioredoxin reductase1 (TrxR1) [171]. In mitochondria, Grx2 has been shown to efficiently catalyze (de)glutathionylation of complex I and SOD1 [173,174].

## 5. Peroxiredoxins (Prdx)

Peroxiredoxins (Prdxs) are cysteine-dependent peroxidase enzymes [132,175], whose low Km for H_2_O_2_ (10 μM) and their ubiquity, comprising up to 0.8% of total protein in some animal cells predispose them for reduction H_2_O_2_ [176]. However, they can also reduce peroxynitrite, peroxynitrous acid, and lipid peroxides [177,178]. Their peroxidatic functions overlap with GPx and catalase, and their catalytic efficiency is lower (~10^5^ M^−1^ s^−1^) compared to GPx (~10^8^ M^−1^ s^−1^) and catalase (~10^6^ M^−1^ s^−1^) [179]. Furthermore, comparing Prdx Km for H_2_O_2_ with that of GPx and catalase exceeding even the millimolar range [180] suggests that the role of Prdx is rather as a sensor of H_2_O_2_ [178] than oxidative stress condition reversal.

Prdxs are divided into the subgroups Prdx1/AhpC, Prdx5, Prdx6, Tpx (thiol peroxidase), PrdxQ/BCP, and AhpE. Human Prdxs can be posttranscriptionally modified by glutathionylation, acetylation, ubiquitination, oxidation (RSOH, RSSR, RSO_2_, RSO_3_), S-nitrosylation, phosphorylation [181] or tyrosine nitration [182]. Prdxs proceed the same catalytic cycle, where the active site cysteine (peroxidatic cysteine, Cys_P_) reduces peroxides and forms Cys_P_-sulfenic acid (RSOH), releasing water or the corresponding alcohol. Some Prdxs contain a second, so-called resolving cysteine (Cys_R_), which reacts with RSOH forming disulfide (Cys_P_-S-S-Cys_R_) and water [183]. Cys_R_ can originate from the adjacent monomer, the same monomer, glutathione, or a redox-relay binding partner [184]. Accordingly, six human Prdxs isoforms are diversified into three subgroups.

In general, the Prdx1 subfamily enzymes are the most highly expressed, making up 0.1–1% of the soluble protein in the cell. The “typical 2-Cys” Prdxs are homodimers with two active sites (having both a Cys_P_ and Cys_R_). The disulfide bond is formed between the two subunits in the reaction of RSOH and Cys_R_ of the other subunit. Reduction of disulfide bond is catalyzed by Trx (Figure 3), tryparedoxin, or alkyl hydroperoxide reductase [179,185]. In the reduced state, PrdxI, II, and IV form decamers or dodecamers such as PrdxIII [186]. Reduced decamers show efficient peroxidase activity and, depending on other posttranslational modifications, form high molecular weight oligomers associated with cell cycle checkpoints, chaperones, and various intracellular processes [187,188,189]. The “atypical 2-Cys” Prdxs (Prdx5) are monomers forming intramolecular disulfide since both Cys_R_ and Cys_P_ are within the same molecule; their reduction is achieved by Trx. The “atypical 2-Cys” Prdxs can form dimers independently of the redox state [179]. The “1-Cys“ Prdxs (Prdx6) contains only Cys_P_ in the *N*-terminus [190]. The resolving electron donor thiol can be glutathione, allowing the formation of a mixed disulfide, while the second donor thiol enables the reduction of the formed disulfide bonding. Ascorbate, lipoic acid, and cyclophilin, but most commonly GSH, can serve as electron donors for disulfide reduction [179,185,191]. Prdx6 reduces phospholipid hydroperoxides using GSH, and also the GST P1-1 class showed the ability to act as phospholipase A_2_ [192]. Hyperoxidation, formation of RSO_2_H or RSO_3_H, and phosphorylation regulate the activity of Prdxs [181]. The “1-Cys” Prdxs are resistant to hyperoxidation. Hyperoxidation can be repaired by sulfiredoxin, but not in human Prdx6 [193].

## 6. Glutathione-S-Transferases (GST)

GSTs belong to the Phase II biotransformation enzymes catalyzing the GSH-mediated peroxide reduction [194] and conjugation of GSH with a variety of reactive electrophiles, most commonly generated by cytochrome P450 metabolism [195]. GSTs expressed ubiquitously, but tissue-specific distribution is probably an adaptive response against endo- and exogenous metabolites [196]. GSTs comprise two distinct superfamilies, membrane-bound microsomal and soluble cytosolic. In humans, cytosolic GSTs are encoded by 16 genes, while the microsomal, at least by six genes, in addition to significant genetic polymorphisms [197]. According to the degree of sequence identity and localization, the cytosolic GSTs (cGSTs) are divided into alpha, mu, pi, omega, theta, delta, sigma, and zeta (A, M, P, O, T, D, S, Z) classes. Mitochondrial GSTs (mGSTs) are divided into A, M, P, and kappa (K) classes. A novel superfamily designated MAPEG (Membrane Associated Proteins in Eicosanoid and Glutathione metabolism) includes members of widespread origin with diversified biological functions. Members of this family are leukotriene C-4 synthase, 5-lipoxygenase activating protein, prostaglandin E synthase, and microsomal glutathione S-transferases (MGST) 1, 2 and 3 [198,199].

Due to polymorphisms, gene duplication, and genetic recombination, GSTs have multiple isoenzymes with overlapping substrate specificity and diversity [200]. In humans, the highest cytosolic GST activity level is present in the liver, whereas the kidney, lung, and intestine show lower activity levels than that of the liver at 22, 66, and 63%, respectively [201]. Intracellularly, some specific GST activities also were detected in the plasma membrane, outer mitochondrial membrane, and nucleus [198]. 

In mammals, GSTs exist as homodimers with analogous tertiary structures [202]. All GSTs have a basic protein fold comprising two subunits with C-terminal and N-terminal domains. The N-terminal domain includes a thioredoxin-like fold, β-α-β-α-β-β-α, where β-β-α motif, known as G-site, serves as the binding site for GSH through the γ-glutamyl unit. The C-terminal domain is diverging [202,203,204]. The conserved proline residue at the N-terminal β3 strand ensures catalytic function and stability of thioredoxin-like proteins [205]. The G-site sequence similarity divides GST into two subgroups. Tyrosine-type GSTs contain Tyr residue (T- or P-class), which activates GSH [206]. Replacement of Tyr by Phe reduces the catalytic activity [207]. The Ser/Cys-type GSTs (O-class) used Ser or Cys to form mixed disulfides with GSH. These GSTs are more involved in redox reactions [208]. Selectivity for the substrates is determined by high variations in hydrophobic amino acid residues in the cleft between domains, called the H-site [208]. 

GSTs transfer GSH to several various electrophilic compounds [209]. The reactions with some compounds, such as benzyl and phenethyl isothiocyanates and alkyl dihalides, can be reversible, increasing their toxicity [210]. Some classes conjugate GSH with epoxides and catalyze isomerization or reduction of harmful peroxides [52]. It was shown that the physiological function of Z-class GSTs is the cis-trans isomerization of 4-maleylacetoacetate to 4-fumarylacetoacetate [211]. The A-class GSTs display selenium-independent GPx activity, thereby reducing phospholipid peroxides and cholesterol hydroperoxides within the membrane without phospholipase A_2_-mediated release [212]. Anionic A-class GSTs also efficiently conjugate 4-hydroxynonenal, balancing lipid production and peroxidation [213]. Furthermore, isomerization of the double bond in selected 3-oxo-Δ^5^-steroids releasing 3-oxo-Δ^4^-steroids has been detected in some A-class GSTs [214]. S-class GSTs enable anti-, proinflammatory, and immunomodulatory functions [215]. From this class, prostaglandin-D_2_ synthase and prostaglandin-E_2_ synthase catalyze the cleavage of prostaglandin H_2,_ forming prostaglandin-D_2_ or E_2_ [216]. The enzyme leukotriene-C_4_ synthase (MAPEG) catalyzes the conjugation of GSH with epoxide leukotriene A_4_ [217]. Unique blood-barrier functions were described for M-class GSTs in the testis and brain [218]. O-class GSTs were able to modulate ryanodine receptor calcium release channels in cardiac muscle due to structural similarities to Chloride Intracellular Channel Proteins (CLIC) [219]. Approximately 15% sequence identity was found between O-class GSTs and CLIC1 [219]. CLIC proteins contain Grx-like active site motiv, Cys-Pro-(Phe/Ser)-(Ser/Cys), present also in O-class GSTs [208,220,221]. CLIC, however, bind GSH covalently creating a mixed disulfide, unlike classical GSTs, which bind GSH in the active site non-covalently but with high affinity [220]. Finally, as indicated by the structural similarity, (de)glutathionylation activity by Menon and Board [222] but also dehydroascorbate reductase, S-(phenylacyl)glutathione reductase [223,224] activities of GSTO1-1 were confirmed. In P-class GSTs, chaperone functions and the influence of the MAPK pathway through JNK and TRAF2 modulation in response to oxidative/nitrosative stress were also detected [225]. One of the unwanted consequences and the subject of intensive ongoing research is resistance to drugs owing to increased GSTs activities [226].

## 7. Glyoxylases (Glo)

The glyoxalase system is a ubiquitous enzymatic network present in the cytoplasm, and some of them are also in the nucleus. It consists of glyoxalase 1 (Glo-1), glyoxalase 2 (Glo-2), and reduced glutathione (GSH) (Figure 2), which perform an essential metabolic function in cells by detoxifying methylglyoxal (MG) and other endogenous harmful metabolites into non-toxic d-lactate [227,228]. As discussed in Rabbani et al. [229], in mammals, methylglyoxal arises in 0.05–0.1% as a minor product from (a) glyceraldehyde-3-phosphate and dihydroxyacetone phosphate degradation in glycolysis, (b) oxidation of acetone by cytochrome P450, (c) oxidation of aminoacetone by semicarbazide amine oxidase, and (d) degradation of glycated proteins and monosaccharides. Methylglyoxal, whose formation can reach 3 mg/kg body weight/day [230], is a glycating agent, forming mainly arginine-derived hydroimidazolone adducts, DNA adducts, and isomeric imidazopurinones [231]. In the glyoxalase system, the rate-limiting enzyme is glyoxalase 1 (Glo-1, lactoylglutathione lyase). Methylglyoxal undergoes spontaneous thiolation with GSH, followed by the Glo-1 catalyzed conversion of methylglyoxal thioacetal to (S)-lactoylglutathione [232,233]. Studies have revealed that Glo-1 is a dimeric metal ion-dependent isomerase converting various glutathione-hemithioacetals to glutathione thioesters [234]. The activity of Glo-1 can be modified by phosphorylation or nitrosylation. While acetylation and oxidation have no effect, acylation of GSH inhibits Glo-1 activity [235]. Glo-2 is a thioesterase catalyzing the hydrolysis of (S)-lactoylglutathione to d-lactate and GSH. Glo-2 predominantly interacts with glutathione moieties allowing hydrolysis of a variety of glutathione substrates [234,236,237]. Glo-3, found in bacteria, catalyzes the conversion of methylglyoxal to d-lactate without the participation of GSH. DJ-1 and its homologs may display this function in humans [232].

Dicarbonyl stress causes protein modification and misfolding, affecting their structure and function, increasing the importance of Glo-1 in detoxification and its implication in the pathophysiology of diseases [238]. Moreover, there is evidence that the Glo-1 gene is a hotspot for copy-number variation associated with multidrug resistance in tumor chemotherapy [239].

## 8. Conclusions

Glutathione reaches the highest concentration in cells, with the predominant component being the reduced form. An electrochemical potential of a redox couple GSH/GSSG at different pH within cell compartments allows reversibility of oxidation or reduction reactions, thereby mediating a cell redox signaling mechanism. Several enzymes use glutathione in reaction mechanisms and fulfill a variety of protective, defensive, synthetic, or signaling roles in cellular metabolism. Either it can be through redox reaction in reduction of peroxides by thiol peroxidases or most common reversible modification, S-glutathionylation by thiol transferases or in conjugation reactions of toxic metabolites through glyoxalase or a variety of other compounds by glutathione-S-transferases. It also raises the question of the suggested genetic basis for differences in glutathione levels. Glutathione is undoubtedly part of a vast complex of cellular machinery processes. Therefore, monitoring it as a marker of specific conditions and dynamic changes in its concentration but also in some systems of which it is a part has a significant value.

## Figures and Tables

**Figure 1 molecules-28-01447-f001:**
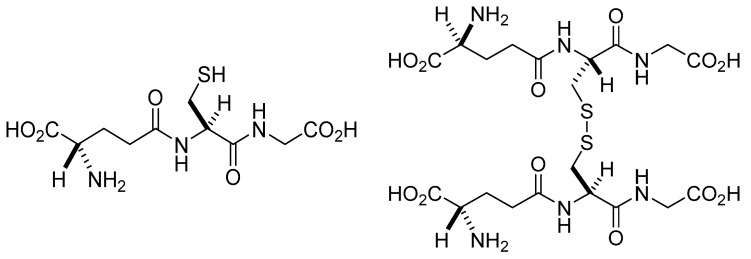
Structure of reduced (GSH) and oxidized (GSSG) forms of glutathione.

**Figure 2 molecules-28-01447-f002:**
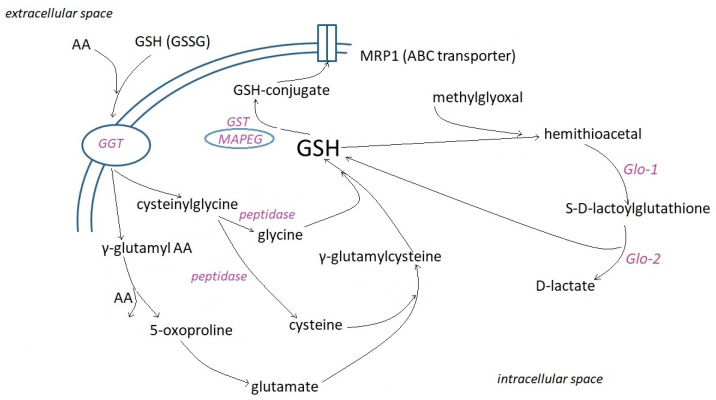
Involvement of γ-glutamyl transpeptidase (GGT), glutathione-S-transferases (GST), their subfamily of Membrane Associated Proteins in Eicosanoid and Glutathione metabolism (MAPEG), and glyoxylases (Glo) in the intracellular metabolism of GSH. MRP1 (multidrug resistance-associated protein 1) transporter facilitates the unidirectional transport of conjugates.

**Figure 3 molecules-28-01447-f003:**
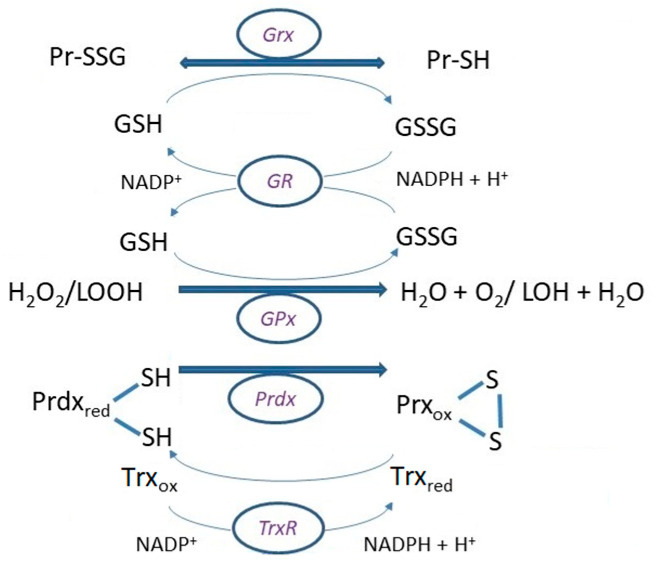
Basic reaction mechanisms of glutathione peroxidase (GPx), and glutaredoxin in (de)glutathionylation using (GSH)GSSG, respectively, and reduction of GSSG by the activity of glutathione reductase (GR) with reducing the power of NADPH + H^+^. Reduction of peroxiredoxin (Prdx) after disposal of peroxides is ensured by thioredoxin (Trx), which is reduced by consumption of NADPH + H^+^ in catalytic efficiency of thioredoxin reductase (TrxR).

## Data Availability

No new data were created or analyzed in this study. Data sharing is not applicable to this article.

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
