# Peer review of "Glutathione-Related Enzymes and Proteins: A Review"

_molecules, 2023, doi:10.3390/molecules28031447_

Round 1
Reviewer 1 Report
This is review of Glutathione-related enzymes and proteins is topical and provides a sufficiently detailed summary of current knowledge on this topic.
There are however some minor corrections required prior to final acceptance for publication:
1. Sentence line 38-40 in introduction is incomplete and requires editing.
" Furthermore, it is one of the endogenous substances involved in the metabolism of endogenous(egs provided)... and exogenous (egs provided) .... [12].
Metabolism of what - some descriptive term needed?
2. Line 64-65 "The consequence of its deficiency results in increased stress conditions and is the base of pathophysiology of many diseases [15]." Suggest you list some examples of types of diseases here.
3. Line 67 "Synthesis of GSH undergoes in the cytoplasm ...."
Suggest to change the word "undergoes" to "occurs"
4. Section 2.3 Line 123. Complete the information and provide estimate % of GSH in the endoplasmic reticulum.
5. Line 126 "This relays on....." should be changed to "This relies on.... "
6. Lines 312-313 "Grx-catalysed (de)glutathionylation is an important event in signal transductions, and it is the protective mechanism of the irreversible oxidation of cysteine residues [121]." Suggest change to
"Grx-catalysed (de)glutathionylation is an important event in signal transductions, and serves as the main protective mechanism against the irreversible oxidation of cysteine residues [121]."
7. Lines 509-510: "O-class GSTs were able to modulate ryanodine receptor calcium release channels in cardiac muscle due to structural similarities to intracellular chloride channels [201]."
It would be more relevant to discuss the ability of O-GST (de)glutathionylation activity as demonstrated by Deepthi Menon and Philip G. Board in their paper published
J Biol Chem. 2013 Sep 6; 288(36): 25769–25779.A Role for Glutathione Transferase Omega 1 (GSTO1-1) in the Glutathionylation Cycle'
8. Similarly, some mention of the CLIC family of proteins as potential new family of GSH-related enzymes.
Author Response
In particular, I would like to thank you for carefully reading the paper, which I highly appreciate. In the following answer, I bring a more detailed description, where the text was adjusted according to the criticized shortcomings.
Once again, I would like to thank you for your comments and hope the modifications helped improve the work's credibility.
Sincerely,
Vaškova
1. Sentence lines 38-40 in the introduction are incomplete and require editing.
" Furthermore, it is one of the endogenous substances involved in the metabolism of endogenous(egs provided)... and exogenous (egs provided) .... [12]. Metabolism of what - some descriptive term needed?
We added the word "compounds."
2. Line 64-65 "The consequence of its deficiency results in increased stress conditions and is the base of the pathophysiology of many diseases [15]." Suggest you list some examples of types of diseases here.
We added examples of diseases according to suggestions.
3. Line 67, "Synthesis of GSH undergoes in the cytoplasm ...." Suggest to change the word "undergoes" to "occurs."
We adjusted the text according to your suggestion.
4. Section 2.3 Line 123. Complete the information and provide an estimated % of GSH in the endoplasmic reticulum.
We have revised and provided relevant information.
5. Line 126, "This relays on....." should be changed to "This relies on.... "
We adjusted the word.
6. Lines 312-313 "Grx-catalysed (de)glutathionylation is an important event in signal transductions, and it is the protective mechanism of the irreversible oxidation of cysteine residues [121]." Suggest a change to "Grx-catalysed (de)glutathionylation is an important event in signal transductions and serves as the main protective mechanism against the irreversible oxidation of cysteine residues [121]."
We have changed the sentence according to the suggestion.
7. Lines 509-510: "O-class GSTs were able to modulate ryanodine receptor calcium release channels in cardiac muscle due to structural similarities to intracellular chloride channels [201]." It would be more relevant to discuss the ability of O-GST (de)glutathionylation activity as demonstrated by Deepthi Menon and Philip G. Board in their paper published in J Biol Chem. 2013 Sep 6; 288(36): 25769–25779.A Role for Glutathione Transferase Omega 1 (GSTO1-1) in the Glutathionylation Cycle'
We added relevant information to meet the reviewer's suggestions.
8. Similarly, some mention the CLIC family of proteins as a potential new family of GSH-related enzymes.
We added and discussed relevant information.
Reviewer 2 Report
Here authors extensively review the role of enzymes and proteins involved in the turnover of GSH, identifying elements associated to redox activity, added to several other functional properties of the antioxidant tripeptide - protective, defensive, synthetic, or signaling roles in cellular metabolism.
Although the cellular compartmentalization (cytosol, mitochondria, endoplasmic reticulum) is mentioned, authors did not mention physiological compartments, in particular to the central nervous system (neurons and glia, please mention the classical work of (Rice and Russo-Menna, 1998), comparing GSH and ascorbate in the neuron-glia compartment as essential antioxidants
I also have not seen any data related to signaling properties of GSH, what is important as an alternative UpToDate to the antioxidant properties. GSH and GSSG have been studied. Authors could devote a few paragraphs to review the molecular mechanisms of extracellular and intracellular targets, involving calcium, transporters, receptors, and enzymes.
Authors should also include recent references regarding pathological involvement of GSH to cancer (Jyotsana et al., 2022; Lee and Roh, 2022; Liu et al., 2021).
The text is well-written, I recommend a small correction in line 38
Furthermore, it is one of the endogenous KEY substances involved in the metabolism of endogenous (e.g., estrogens, leukotrienes, prostaglandins) and exogenous (e.g., drugs, non-energy-producing xenobiotics) COMPOUNDS.
Jyotsana, N., Ta, K.T., DelGiorno, K.E., 2022. The Role of Cystine/Glutamate Antiporter SLC7A11/xCT in the Pathophysiology of Cancer. Front Oncol. 12, 858462.
Lee, J., Roh, J.L., 2022. SLC7A11 as a Gateway of Metabolic Perturbation and Ferroptosis Vulnerability in Cancer. Antioxidants (Basel). 11.
Liu, X., et al., 2021. NADPH debt drives redox bankruptcy: SLC7A11/xCT-mediated cystine uptake as a double-edged sword in cellular redox regulation. Genes Dis. 8, 731-745.
Rice, M.E., Russo-Menna, I., 1998. Differential compartmentalization of brain ascorbate and glutathione between neurons and glia. Neuroscience. 82, 1213-23.
Author Response
In particular, I would like to thank you for carefully reading the paper, which I highly appreciate. In the following answer, I bring a more detailed description, where the text was adjusted according to the criticized shortcomings.
Once again, I would like to thank you for your comments and hope the modifications helped improve the work's credibility.
Sincerely,
Here authors extensively review the role of enzymes and proteins involved in the turnover of GSH, identifying elements associated to redox activity, added to several other functional properties of the antioxidant tripeptide - protective, defensive, synthetic, or signaling roles in cellular metabolism.
Although the cellular compartmentalization (cytosol, mitochondria, endoplasmic reticulum) is mentioned, authors did not mention physiological compartments, in particular to the central nervous system (neurons and glia, please mention the classical work of (Rice and Russo-Menna, 1998), comparing GSH and ascorbate in the neuron-glia compartment as essential antioxidants
I also have not seen any data related to signaling properties of GSH, what is important as an alternative UpToDate to the antioxidant properties. GSH and GSSG have been studied. Authors could devote a few paragraphs to review the molecular mechanisms of extracellular and intracellular targets, involving calcium, transporters, receptors, and enzymes.
Authors should also include recent references regarding the pathological involvement of GSH in cancer (Jyotsana et al., 2022; Lee and Roh, 2022; Liu et al., 2021).
We included a paragraphs according to the reviewer's suggestion.
The text is well-written, I recommend a small correction in line 38
Furthermore, it is one of the endogenous KEY substances involved in the metabolism of endogenous (e.g., estrogens, leukotrienes, prostaglandins) and exogenous (e.g., drugs, non-energy-producing xenobiotics) COMPOUNDS.
We added the word "compounds. "
Reviewer 3 Report
In this manuscript through an overview of the relevant literature authors summarized role of glutathione as well as glutathione-related enzymes and proteins. Even though there are many reviews about glutathione that authors included in References list, authors gave very interesting overview with stress to role of these systems in redox and detoxification reactions. In my opinion this work will advance the literature and be of interest to a broad audience. I suggest only a few revisions prior to publication:
Line 32: double parentheses (Figure 1))
Line 40: at the end of sentence put word - compounds or substances
Line 52: there is mistake – change to 0.25M ionic strength
Line 62: put reference on the end of sentence “.... are closely related to carbohydrate metabolism [ref]”.
Line 78-79: change “protein cysteine” to “ cysteine turnover in body protein pool”
Line 97: Please check statement about absent GGT activity in hepatocytes. GGT are usual hepatobiliary marker.
Figure 2: Please revise Figure 2. The figure doesn't show that GSH is hydrolyzed by GGT to γ-glutamyl AA and Cys-Gly and further Cys-Gly is hydrolyzed to Cys and Gly by peptidase. At Figure it seems that GGT forms only γ-glutamyl AA which cleaves to Gly and Cys-Gly. Also in Figure caption abbreviation MRP1is not introduced.
Equation (1): replace = to →
Line 263: replace 52Kd to 52 kD
Line 353: delete the bracket after GSTs and put comma
Line 405: put “[“ in front of Fe-S]
Figure 3. It is not easy to follow. Different mechanisms authors can mark with letters. Also function of GR is not obvious in figure. Prdx or Prx?
Line 416: In Figure caption in line 416 write peroxiredoxin before Prx and put abbreviation in brackets.
Line 427: Prdx or Prx?
Line 443: What do you mean exactly that Trx is tryparedoxin (as membre of thioredoxin) or thioredoxin?
Section -5. Peroxiredoxins: authors don’t refer to the Figure 3 in this section.
Line 454: Prdx6 or Prx6?
Line 469: Pi writes in lowercase letters or other isoforms with capital letters also
Line517: I think that authors mean Figure 2 instead of Figure 3
Finally I think that manuscript is appropriate for publishing after revision by authors.
Author Response
In particular, I would like to thank you for carefully reading the paper, which I highly appreciate. In the following answer, I bring a more detailed description, where the text was adjusted according to the criticized shortcomings.
Once again, I would like to thank you for your comments and hope the modifications helped improve the work's credibility.
Sincerely,
In this manuscript, through an overview of the relevant literature, authors summarized the role of glutathione as well as glutathione-related enzymes and proteins. Even though there are many reviews about glutathione that authors included in References list, authors gave very interesting overview with stress to the role of these systems in redox and detoxification reactions. In my opinion, this work will advance the literature and be of interest to a broad audience. I suggest only a few revisions before publication:
Line 32: double parentheses (Figure 1))
Line 40: at the end of the sentence, put the word - compounds or substances
We removed double parenthesis and added the word "compounds".
Line 52: there is a mistake – change to 0.25M ionic strength
We corrected "0.25".
Line 62: put reference on the end of a sentence ".... are closely related to carbohydrate metabolism [ref]".
We added relevant references.
Line 78-79: change "protein cysteine" to "cysteine turnover in body protein pool
We have changed the sentence according to the suggestion.
Line 97: Please check the statement about absent GGT activity in hepatocytes. GGT is a usual hepatobiliary marker.
The activity of GGT in the liver under physiological conditions is very low. It rises under conditions when its synthesis is disturbed, and the level of glutathione in the liver becomes depleted, which is usually also associated with disrupting hepatocyte functions. Therefore, GGT becomes an ideal marker of damage. Sufficient synthesis and amount of glutathione in the liver are physiologically ensured by mechanisms explained below, which we also mentioned in Diverse effects of glutathione in the pathophysiology of some organ diseases. In: Glutathione: Biosynthesis, Functions and Biological Implications, 219, pp. 169-188.
The main mechanism of hepatocyte GSH turnover is its efflux into bile and blood, as no significant activity of GGT was determined in the liver [Yale Journal of Biology and Medicine 1981, 54:497-502]. GSH thus serves to supply its synthetic precursors to other tissues that express GGT, particularly the kidney and intestine. However, the fact is that the disruption of GSH synthesis in the liver has an overall impact on GSH homeostasis in the body. The decrease of GSH synthesis is the basis of all pathogenesis of liver disorders such as cholestasis, endotoxemia, and alcoholic and non-alcoholic fatty liver diseases [Biochimica et Biophysica Acta, 2013, 1830:3143–53]. In addition, systemic oxidative stress arising during liver disease can also cause damage to extra-hepatic organs such as the brain and kidney. Systemic oxidative stress was suggested as a significant "first hit", acting synergistically with ammonia to induce brain edema in chronic liver failure [Free Radical Biology and Medicine, 2012, 52:1228–35; Kidney International, 2015, 87:948–62]. Due to the lack of GGT, ensuring a sufficient concentration of GSH for redox, conjugation reactions, and other tissue supply in the liver is possible only by synthesizing and reducing oxidized glutathione via glutathione reductase.
In hepatocytes, a series of processes are necessary for cysteine availability, such as membrane transport of cysteine via the ASC system (Alanine Serine Cysteine system), cysteine via the xc− system (glutamate/cysteine antiporter), which is induced under oxidative stress, and methionine via the L system (leucine-preferring system) [Journal of Membrane Biology, 1986, 89:1–8]. While the liver catabolizes up to half the daily intake of methionine, and the transsulfuration (cystathionine) pathway is particularly active in hepatocytes, there is little or no evidence of transsulfuration outside the liver.
Figure 2: Please revise Figure 2. The figure doesn't show that GSH is hydrolyzed by GGT to γ-glutamyl AA and Cys-Gly and further Cys-Gly is hydrolyzed to Cys and Gly by peptidase. At Figure it seems that GGT forms only γ-glutamyl AA which cleaves to Gly and Cys-Gly. Also in the Figure caption abbreviation MRP1 is not introduced.
Equation (1): replace = to →
We have revised Figure 2, its caption, and mentioned equation.
Line 263: replace 52Kd with 52 kD
Line 353: delete the bracket after GSTs and put a comma
Line 405: put "["in front of Fe-S]
We made corrections according to suggestions.
Figure 3. It is not easy to follow. Different mechanisms authors can mark with letters. Also function of GR is not obvious in figure. Prdx or Prx?
Line 416: In Figure caption in line 416 write peroxiredoxin before Prx and put abbreviation in brackets.
Line 427: Prdx or Prx?
We have revised Figure 3, and we unified the abbreviations.
Line 443: What do you mean exactly that Trx is tryparedoxin (as member of thioredoxin) or thioredoxin?
We explained the abbreviation in the text.
Section -5. Peroxiredoxins: authors don't refer to Figure 3 in this section.
Line 454: Prdx6 or Prx6?
We mentioned Figure 3 within section 5, and we unified the abbreviations.
Line 469: Pi writes in lowercase letters or other isoforms with capital letters, also
Line517: I think that authors mean Figure 2 instead of Figure 3.
We made corrections according to suggestions.